# Dual-color fluorescent nanoparticles showing perfect color-specific photoswitching for bioimaging and super-resolution microscopy

Dojin Kim [1], Keunsoo Jeong [2], Ji Eon Kwon [1], Hyeonjong Park[2], Seokyung Lee[2], Sehoon Kim [2,3] & Soo Young Park[1]

Dual-emissive systems showing color-specific photoswitching are promising in bioimaging and super-resolution microscopy. However, their switching efficiency has been limited because a delicate manipulation of all the energy transfer crosstalks in the systems is unfeasible. Here, we report a perfect color-specific photoswitching, which is rationally designed by combining the complete off-to-on fluorescence switching capability of a fluorescent photochromic diarylethene and the frustrated energy transfer to the other fluorescent dye based on the excited-state intramolecular proton transfer (ESIPT) process. Upon alternation of UV and visible light irradiations, the system achieves 100% switching on/off of blue emission from the diarylethene while orange emission from the ESIPT dye is unchanged in the polymer film. By fabricating this system into biocompatible polymer nanoparticles, we demonstrate microscopic imaging of RAW264.7 macrophage cells with reversible blue-color specific fluorescence switching that enables super-resolution imaging with a resolution of 70 nm.

[1] Center for Supramolecular Optoelectronic Materials, Research Institute of Advanced Materials (RIAM), Department of Materials Science and Engineering, Seoul National University, 1 Gwanak-ro, Gwanak-gu, Seoul 08826, Korea. [2] Center for Theragnosis, Korea Institute of Science and Technology, Hwarangno 14-gil 5, Seongbuk-gu, Seoul 02792, Korea. [3] KU-KIST Graduate School of Converging Science and Technology, Korea University, Seongbuk-gu, Seoul 02841, Korea. Correspondence and requests for materials should be addressed to S.Y.P. (email: parksy@snu.ac.kr)

Photoswitchable nanoparticles, whose fluorescence emission can be turned on/off reversibly by light irradiations, have attracted great attention because of their potential in fluorescence imaging[1–7]. For instance, they can selectively highlight bio-systems by eliminating autofluorescence, and also allow for super-resolution microscopy such as PALM (photoactivation localization microscopy), STORM (stochastic optical reconstruction microscopy), and RESOLFT (reversible saturable optical linear fluorescence transitions)[8–16], whose fundamental principles rely on optical switching[17–20]. For these purposes, a variety of photoswitchable nanoparticles have been reported so far; however, most of them are single-emissive systems and capitalize on simple one-color photoswitching[21–31]. Such single-color photoswitching probes have a critical drawback in that their location is lost in the switched-off state when they are moving in bio-systems, such as living cells.

Color-specific photoswitchable nanoparticles with dual-color fluorescence emission have emerged as a promising probe to address the above issue, in which only one emission color is switched on/off reversibly upon light irradiations, while the other color lights up the nanoparticles invariably even when the former is in the switched-off state[32–39]. In the design of such color-specific photoswitchable nanoparticles, a general scheme has been a three-component system comprising two fluorophores with different emission colors and a photochromic dye as depicted in Fig. 1a. At the colorless photochrome state with a high $S_1$ energy, both fluorophores do not undergo the fluorescence quenching interaction with the photochromic dye. If the mutual energy transfer (ET) between the fluorophores (ET A in Fig. 1a) can be blocked, they would emit their own fluorescence colors independently. Upon photoswitching by UV irradiation, the colored photochrome state with a low $S_1$ energy is generated and triggers energy transfers for fluorescence quenching (ET B in Fig. 1a). In Fig. 1a, the photochromic dye turns off the fluorescence only from Fluorophore 1 whose exciton energy is higher than that of the colored photochrome state. In the meantime, Fluorophore 2 would remain emissive invariably under the condition where the ET A is frustrated, to render photoswitching of the whole dual-emissive system color-specific.

As discussed above in the three-component system, color-specific photoswitching capitalizes on elaborate engineering of energy transfers among the components. The downhill ET B and the uphill ET C (see Fig. 1a) can be favorably controlled (permitted and blocked, respectively) according to the FRET (Förster resonance energy transfer) principle[40]. However, blocking of the interfluorophore energy transfer (ET A) remains a major

challenge, and several attempts have been made. For example, Hell et al. reported a silica nanoparticle where two different fluorophores are radially separated beyond their Förster distance[41]. Jovin et al. suggested using separate excitation sources to readout individual fluorescence signals from quantum dot nanoparticles[42]. Although both the dual-emissive systems achieved selective single-color photoswitching by blocking the interfluorophore energy transfer, the efficiencies of switching on/ off by diarylethenes, one of the most commonly used photochromic dyes[43,44], were compromised (80 and 50%, respectively) with other problems, such as the limitation of nanoparticle size control, complex fabrication process, and additional cost for special readout instruments.

As an alternative strategy to block the ET A, Branda et al. exploited a lanthanide-doped inorganic nanoparticles[45], where green and red emissions come from two different electronic transitions of the lanthanide dopants and thus inherently do not undergo energy transfer between each other. Upon photoswitching of a diarylethene, only the green emission could be turned on/off selectively, but with a limited efficiency of 54%. Recently, we demonstrated color-specific photoswitching by employing the excited-state intramolecular proton transfer (ESIPT) process in an all-organic system, which consists of two ESIPT dyes with different emission colors (blue and orange) and a diarylethene[46]. As the energy transfer between the ESIPT dyes was blocked according to the proton transfer morphing[47], the blue emission could be selectively switched (59%) upon photochromic switching.

Although all systems described above nicely demonstrated different principles of blocking ET A for attaining color-specific photoswitching, the efficiencies of switching on/off were limited due to the inefficient process of fluorescence quenching by ET B (see Fig. 1a). Indeed, in such three-component systems (i.e., two fluorophores and a photochromic switch for fluorescence quenching), it is extremely hard to elaborately control energy transfers to the diarylethene from fluorophores in a way that the energy transfer from the high-energy fluorophore (ET B) is progressed perfectly while completely avoiding that from the low-energy one (ET C). This is because their absorption/emission spectra are quite broad due to the existence of many vibrational modes as typically observed in organic molecules, which generally causes mutual interferences compromising the color-specificity of energy transfer[40].

Our aim in this work is to develop an original strategy for the perfect color-specific photoswitching, in which the high-energy

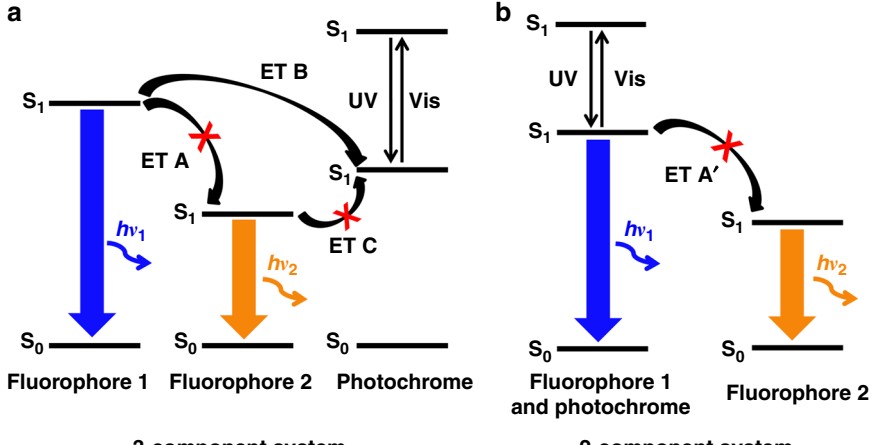

**Fig. 1** Energy transfer schemes for color-specific photoswitching. Schematic illustrations of the energy levels and the energy transfer (ET) processes upon UV and visible-light irradiation in color-specific photoswitching systems: (**a**) three-component system and (**b**) two-component system

blue emission is selectively and completely switched on/off with a 100% efficiency while the low-energy orange emission is unaltered. Here, we present a two-component system showing perfect color-specific photoswitching, which is composed of a blue fluorescent turn-on diarylethene and an orange-fluorescent ESIPT dye. The turn-on diarylethene is chosen as an energy donor not only because it can simplify the whole system (only one energy transfer process (**ET A′**) left to be considered, as depicted in Fig. 1b), but also because the fluorescence switching is governed no more by the intermolecular energy transfer but by the unimolecular photochromic reaction of the turn-on diarylethene itself, which enables 100% emission switching between the emissive and nonemissive states. As an innovative way to block the **ET A′**, we employed an orange-emitting ESIPT dye whose absorption has no spectral overlap with the photochromic blue emission due to its peculiarly large Stokes shift. By fabricating this two-component dual-emissive system into biocompatible polymer nanoparticles, we demonstrate perfect color-specific photoswitching in cells, which allows for super-resolution microscopy under a PALM setup.

## Results

**Materials**. Figure 2a and b show two organic molecular components that constitute our perfect color-specific photoswitching system, i.e., 3,3′-(perfluorocyclopent-1-ene-1,2-diyl)bis(2-ethyl-benzo[b]thiophene 1,1-dioxide) (DBTEO) as a photoswitchable blue emitter and 3-(1-phenyl-1H-phenanthro[9,10-d]imidazol-2-yl)naphthalen-2-ol (HPNIC) as an orange emitter, respectively. DBTEO is a photochromic diarylethene derivative bearing sulfone moieties (Fig. 2a), which is known to undergo bistable photochromic reactions upon UV/vis light irradiations between an open form (**O**) and a closed form (**C**), where the **C** form emits strong blue fluorescence[48,49]. Therefore, upon UV-light irradiation, the fluorescence can be switched on from the initial dark state (the nonfluorescent **O** form); i.e., the fluorescence on/off ratio is close to infinity. On the other hand, HPNIC is an imidazole-based ESIPT molecule[50,51]. As shown in Fig. 2b, when HPNIC absorbs light, it undergoes a characteristic four-level tautomerization cycle (**E → E\* → K\* → K → E**). In the cycle, there are two different species of the molecule (enol/keto tautomer). Due to the fast structural morphing by proton transfers, HPNIC exists in the enol (**E**) form in the ground state, but in the keto (**K**) form in the excited state. Therefore, light is absorbed by the ground-state enol form (**E → E\***), but the fluorescence is emitted by the excited-state keto form (**K\* → K**), which features an abnormally large Stokes shift with no overlap between its absorption and emission spectra. As a result, the absorption band of orange-fluorescent HPNIC is confined within UV-light region, which is in sharp contrast to visible absorption bands of other common orange-fluorescent dyes.

**Photophysical studies in PMMA films**. To verify the unique photophysical/photochemical properties, the two dyes were doped in poly(methyl methacrylate) (PMMA) films and subjected to the measurement of UV–vis absorption and photoluminescence (PL) spectra. As shown in Fig. 3a, DBTEO in the **O** form has an absorption band only in the UV-light region ($\lambda_{max,abs,O} = 311$ nm). Upon UV-light irradiation, the photocyclization reaction occurred to produce the **C** form, and accordingly the DBTEO film reached to a photostationary state (PSS). In the PSS, a new absorption peak appeared in the visible-light region ($\lambda_{max,abs,C} = 415$ nm), which is attributed to the **C** form of DBTEO. Because both the isomers are thermally stable against photochromic reaction (see Fig. 2a), the two absorption bands are interconverted only by UV/vis light irradiations.

Interestingly, the **C** form of DBTEO emits an intense sky-blue fluorescence ($\lambda_{max,em,C} = 498$ nm, $\Phi_F = 0.55$), while the **O** form is virtually nonfluorescent. It is noteworthy that visible-light irradiation to the film converted all the **C** form of DBTEO molecules into the initial nonemissive **O** form because only the **C** form can be excited by the visible light. This is the main reason why we can perfectly switch the blue emission of the film between

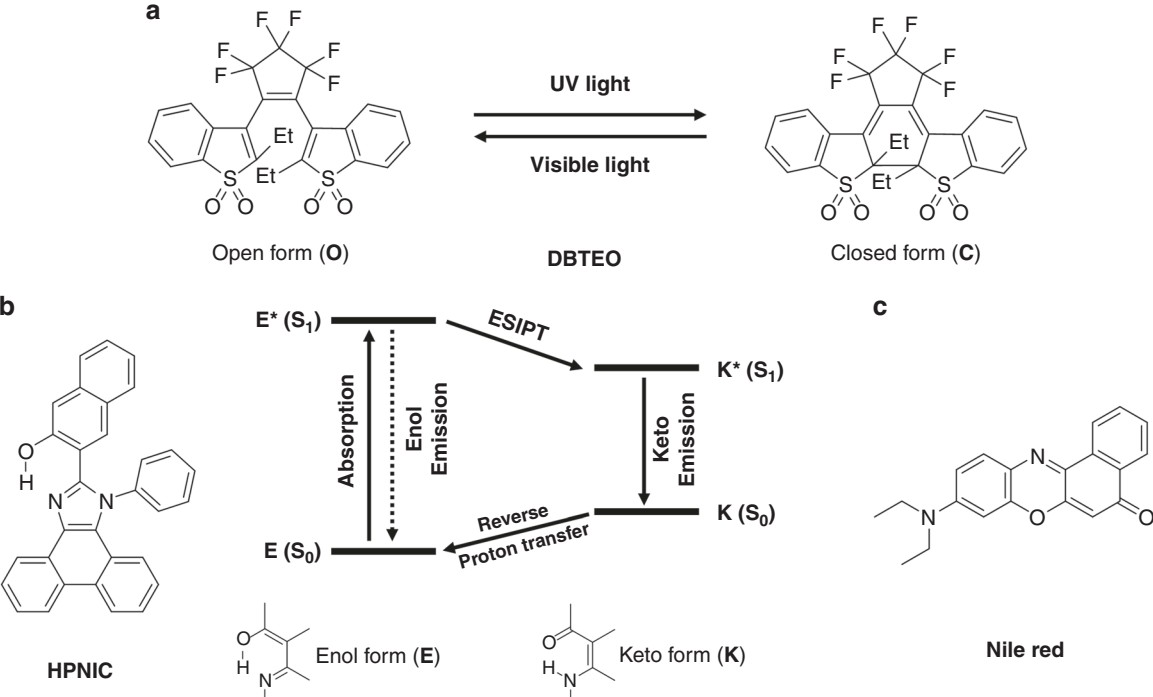

**Fig. 2** Molecular structures and photochemical reactions. **a** Chemical structure and photochromic reaction of DBTEO. **b** Chemical structure of HPNIC and schematic illustration of the ESIPT process. **c** Chemical structure of Nile Red

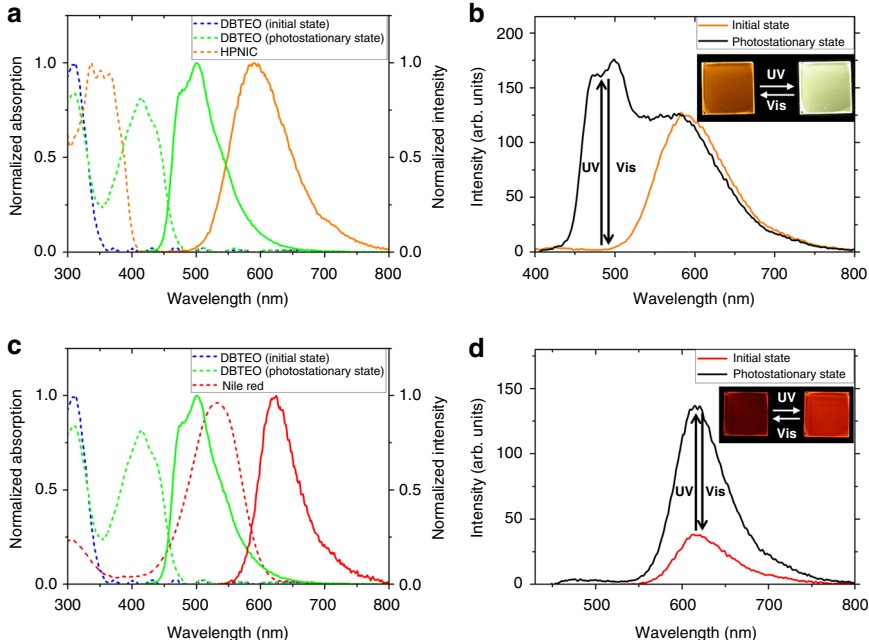

**Fig. 3** Absorption and emission properties in polymer films. **a** Normalized absorption (dashed lines) and photoluminescence (PL) (solid lines) spectra of DBTEO and HPNIC (5 wt% doped in PMMA). **b** PL spectra changes of the mixture film containing DBTEO and HPNIC (DBTEO:HPNIC = 1.5 wt%:1 wt% doped in PMMA) upon UV and visible-light irradiation. The excitation wavelength of PL spectra was 365 nm. **c** Normalized absorption (dashed lines) and PL (solid lines) spectra of DBTEO and Nile Red (5 wt% doped in PMMA). **d** PL spectra changes of the mixture film containing DBTEO and Nile Red (DBTEO:Nile Red = 1.5 wt%:1 wt% doped in PMMA) upon UV and visible-light irradiations. The excitation wavelength of PL spectra was 365 nm. The inset figures in (**b**) and (**d**) are photographs of fluorescence switching upon light irradiations of the PMMA films

on and off state (i.e., the fluorescence on/off ratio is close to infinity) by alternate irradiations of UV and visible lights.

On the other hand, the HPNIC film showed an absorption ($\lambda_{\mathrm{max,abs,E}} = 337$ nm) in the UV region and a large Stokes-shifted orange emission ($\lambda_{\mathrm{max,em,K}} = 590$ nm, $\Phi_{\mathrm{F}} = 0.11$), each of which is attributed to the ground-state **E** form and the excited-state intramolecular proton transferred **K\*** form of HPNIC, respectively. The absence of an emission with normal Stokes shift (**E\*** → **E**) and also no new arising absorption band (**K** → **K\***) imply that the ESIPT as well as the ground-state backward proton transfer occur very fast and efficiently in the HPNIC films (see Fig. 2b). Most importantly, due to the abnormally large Stokes shift, there is no spectral overlap between the emission of DBTEO and the absorption of HPNIC, and vice versa. It is therefore expected that the energy transfer between the two dyes is totally frustrated to achieve color-specific photoswitching when they are mixed in the polymer film.

Indeed, as shown in Fig. 3b, the mixture film containing 1.5 wt% DBTEO (**O** form) and 1 wt% HPNIC initially displayed only an orange HPNIC emission peaking at 590 nm. Upon UV-light irradiation, it was observed that a sky-blue emission of DBTEO (**C** form) peaking at 498 nm increased gradually accompanied by an absorption increase at around 415 nm (Supplementary Fig. 1), resulting in the dual fluorescence emission. Upon subsequent visible-light irradiation (>420 nm), the sky-blue emission was perfectly turned off by the cycloreversion reaction from **C** form to **O** form of DBTEO, while the orange emission was invariant to demonstrate the reversibility of color-specific photoswitching.

To check the occurrence of energy transfer between the two dyes, fluorescence lifetime changes of the two emissions were measured. The **C** form of DBTEO displayed 3.74 ns of fluorescence lifetime (τ) at the PSS in the solitary film (see Supplementary Fig. 2). In the mixture film with HPNIC, it was observed that the lifetime of DBTEO (**C** form) was minimally affected with only a slight decrease (τ = 3.40 ns). Considering the

spectral mismatch discussed in Fig. 3a, it can be reasonably inferred that the energy transfer from the **C** form of DBTEO to HPNIC is highly restricted (i.e., **ET A′** is blocked in Fig. 1b). Most importantly, the emission intensity as well as fluorescence lifetime of the orange emission from HPNIC was virtually unaffected by the photochromic conversion between **O** form and **C** form of DBTEO in the mixture film (see Fig. 3b; Supplementary Fig. 3). This means that the two dye molecules undergo their own photochemical reaction (ESIPT and photochromism) and emit fluorescence independently without significant energy-transferring interferences in the mixture film.

For a control experiment, we prepared another mixture film containing Nile Red (Fig. 2c) as a conventional orange emitter instead of HPNIC. As shown in Fig. 3c, the spectral overlap between the absorption of Nile Red and the emission of the **C** form of DBTEO is very large (spectral overlap integral, $J(\lambda) = 2.05 \times 10^{15}$ M$^{-1}$ cm$^{-1}$ nm$^4$), thus it is anticipated that the energy transfer from the **C** form of DBTEO to Nile Red would be efficient in the mixture film (i.e., **ET A′** is permitted in Fig. 1b). Indeed, the mixture film was shown to be no more dual-emissive; even after UV-light irradiation, the sky-blue emission from the **C** form of DBTEO was totally quenched (~100% quenching efficiency), and only the emission of Nile Red was observed at the PSS of the mixture film. Interestingly, the emission intensity of Nile Red was reversibly changed by the photochromic reaction between **O** form and **C** form of DBTEO (Fig. 3d), indicative of the energy harvesting effect from the emissive **C** form of DBTEO to Nile Red. The fluorescence lifetime of Nile Red was virtually maintained upon photochromic reaction, which is typical behavior of energy acceptors (Supplementary Fig. 4). Meanwhile, the lifetime of DBTEO in the emissive **C** form could not be measured because the emission was totally quenched via the complete energy transfer from DBTEO (**C** form) to Nile Red. These results imply that restricting energy transfer between the two fluorophores (**ET A′**

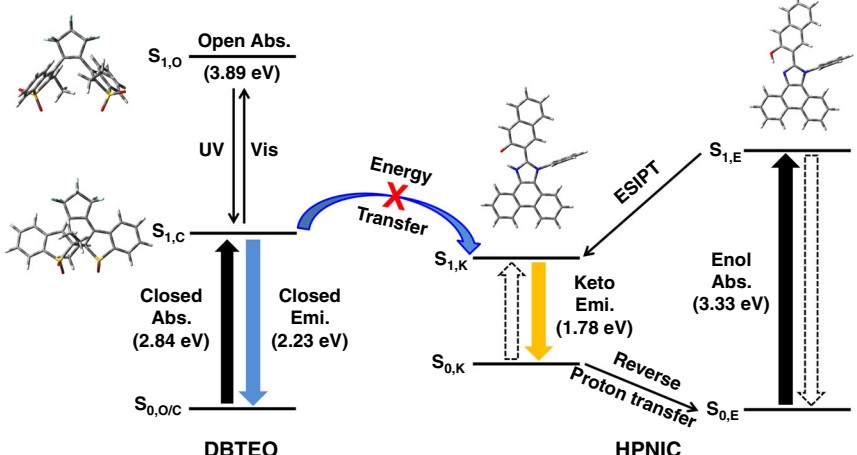

**Fig. 4** Principle of color-specific photoswitching. Schematic illustration of the photochromic reaction of DBTEO, the ESIPT process of HPNIC, and the frustrated ET between DBTEO and HPNIC. The energy levels in the scheme are calculated based on the density functional theory (DFT) methods

in Fig. 1b) is essential for realizing perfect color-specific photoswitching in the dual-emissive system.

**Photophysical principle based on DFT calculation**. To have a deeper insight into the energy-transferring interaction between DBTEO and HPNIC (**ET A′** in Fig. 1b) in the mixture system, density functional theory (DFT) calculations were carried out (see Fig. 4). When DBTEO exists initially in the **O** form, it absorbs UV light (3.89 eV, $S_{0,O} \rightarrow S_{1,O}$) to be excited. In the excited state, it undergoes a photocyclization reaction into the **C** form, which is thermally stable in the ground state and has the lower absorption energy (2.84 eV, $S_{0,C} \rightarrow S_{1,C}$). The emission energy of DBTEO in the **C** form was estimated to be lower in the visible-light region ($S_{1,C} \rightarrow S_{0,C}$, 2.23 eV) by the time-dependent DFT (TD-DFT) calculation. In the case of HPNIC, it exists initially as the **E** form which has the UV-light absorption ($S_{0,E} \rightarrow S_{1,E}$, 3.33 eV) to be excited. In the excited state, photo-induced tautomerization instantaneously occurs to transform the excited HPNIC molecules from the **E** form to the **K** form by transferring the proton from the hydroxyl group to the imidazole ring ($S_{1,E} \rightarrow S_{1,K}$). Then, the excited **K** form relaxes down to the ground state with visible-light emission ($S_{1,K} \rightarrow S_{0,K}$, 1.78 eV). In sharp contrast to the stable **C** form of DBTEO, the **K** form of HPNIC is transient in the ground state, and thus immediately after radiative relaxation, goes back to the original stable form of **E** by a reverse proton transfer reaction ($S_{0,K} \rightarrow S_{0,E}$).

It should be noted that the **E** form of HPNIC (3.33 eV) has much higher absorption energy than the emission energy of DBTEO in the **C** form (2.23 eV), being ruled out as an energy acceptor. The only potential energy acceptor in the system is the **K** form of HPNIC. However, as discussed above, its population is depleted by immediate drainage to the ground-state dominant form of **E** (actually showing no steady state $S_{0,K} \rightarrow S_{1,K}$ absorption). Therefore, the transient nature of the **K** form acceptor also rules the chance of energy transfer out to render the dyes in the mixed system emissive independently, as depicted in Fig. 4.

**Biocompatible PCL nanoparticles**. Having characterized the two-component color-specific photoswitching system, we applied it to bioimaging in a form of biocompatible polymer nanoparticles (NPs). To this end, we fabricated poly(caprolactone) (PCL) NPs containing DBTEO and HPNIC molecules by modified literature procedures[52]. We chose PCL as a polymeric container because it is approved as a biocompatible polymer by the

Food and Drug Administration (FDA) for use in bio-applications, such as drug delivery[53]. In addition, it is anticipated that PCL would provide the fluorophores with surrounding environments (e.g., polarity) similar to that of PMMA due to the structural similarity. Indeed, the photophysical/photochemical properties of DBTEO and HPNIC were observed to be almost identical in the PMMA and PCL films (see Supplementary Fig. 5 and Supplementary Table 1). Notably, these optical properties including absorption, emission, and fluorescence quantum yield were negligibly changed in the PCL NPs, except for some scattering effect in absorption spectra (see Supplementary Figs. 6, 7 and Supplementary Table 1). As schematically illustrated in Fig. 5a, the NPs are composed of a hydrophobic PCL core in which DBTEO and HPNIC are doped, and a hydrophilic surface stabilized with a FDA-approved polymer surfactant (Pluronic F-68). It turned out that the NPs maintained a transparent dispersion in an aqueous environment very stably without aggregation for more than a month. Transmission electron microscopic (TEM) images show that the obtained NPs are spherical in shape and their average size is 26.7 ± 3.9 nm (Fig. 5b). On the other hand, the number-averaged hydrodynamic size of these NPs in water was determined as 57.3 ± 5.6 nm by dynamic light-scattering (DLS) measurements, which is tiny enough for facile delivery into the cells (Fig. 5c)[54].

Figure 6a and Supplementary Fig. 8 show the photochromic and photoluminescence properties of the NPs. Notably, the emission spectra and photoswitching capabilities of the NPs in water are almost identical to those of the mixture in PMMA film. It is well known that the ESIPT mechanism can be interrupted in aqueous conditions due to the intermolecular hydrogen bonding with water molecules that interferes with the intramolecular hydrogen bonds of ESIPT molecules[55]. However, it was found that HPNIC in the PCL NPs retains the characteristic ESIPT behavior with efficient orange keto emission, suggesting that it is well embedded in the polymer matrix and thus isolated from surrounding water molecules (Supplementary Fig. 9). Accordingly, the emission color of the NPs is orange when DBTEO exists in the colorless **O** form (Fig. 6a). Upon UV-light exposure for 30 s, the whole emission color was quickly changed to white, due to the appearance of photochromic blue emission. The reverse reaction could be achieved with ease by visible-light irradiation ($\lambda > 420$ nm) for 2 min, to completely switch back to the blue-free orange emission. It is worth noting that this perfect and reversible color-specific photoswitching of the NPs could be repeated stably by alternating UV and visible-light irradiations (Fig. 6b). The

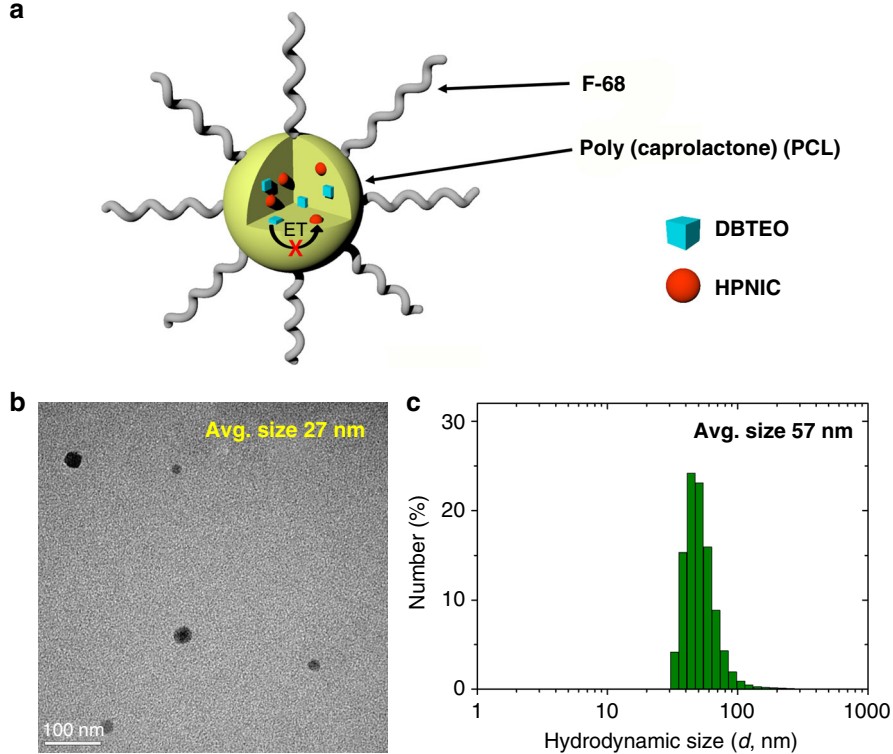

**Fig. 5** Biocompatible nanoparticles. **a** Schematic representation of the PCL nanoparticles (NPs) doped with DBTEO and HPNIC and their photoswitching reaction by UV/visible-light irradiations. **b** Field-emission transmission electron microscopic (FE-TEM) image of the NPs. **c** Size distribution and average size of the NPs in water measured by dynamic light scattering (DLS)

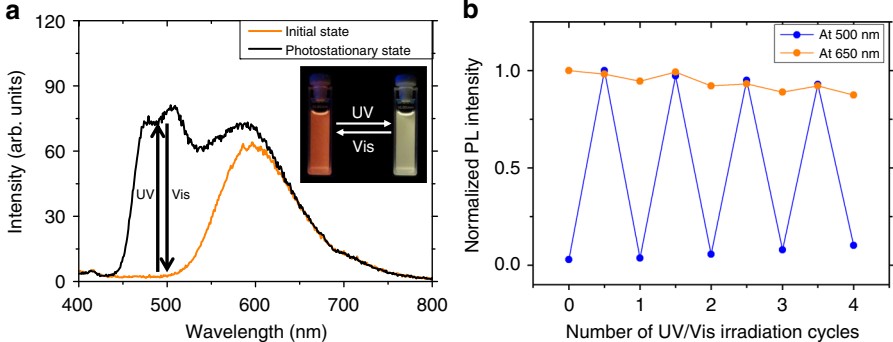

**Fig. 6** Emission properties in nanoparticles. **a** PL spectra changes of the PCL nanoparticle containing DBTEO and HPNIC (DBTEO:HPNIC = 1.5:1 doped in PCL, the total concentration of dyes to PCL matrix is 1 wt%) upon UV and visible-light irradiations. The inset is a photograph of fluorescence switching in the nanoparticle. **b** Reversibility test of the nanoparticle with alternation of UV and visible-light irradiations

photocyclization and photocycloreversion quantum yields of DBTEO in PCL NPs were measured to be 0.20 and 0.083, respectively, in which DBTEO in dioxane solution was used as a reference (Supplementary Figs. 10, 11)[49]. Both photochromic quantum yields of DBTEO in PCL NPs are lower than those (0.28 and 0.18, respectively) in dioxane solution, which is most likely attributed to the rigid environment of the PCL matrix.

**Bioimaging and super-resolution microscopy**. To evaluate the applicability of our two-component color-specific photoswitching system for bioimaging, we explored the in vitro imaging of RAW264.7 macrophage cells by treating them with the NPs in the switched-off state (the **O** form). The fluorescence confocal microscopic images displayed intense and stable orange emission in the cytoplasmic region of cells, well revealing that the intracellular delivery of the NPs could be visualized by the ESIPT

signal of HPNIC even when the blue emission of DBTEO (**C** form) was turned off (Fig. 7). The always-on orange emission presented that the tiny and biocompatible PCL NPs are taken up by cells in a time-dependent manner (Supplementary Fig. 12). Furthermore, upon repetitive UV/vis light irradiations, the blue emission of DBTEO in the **C** form was reversibly turned on and off while the orange emission of HPNIC remained virtually constant, evidencing that the color-specific photoswitching is indeed operative in the intracellular environment owing to the high structural integrity of PCL-based NPs. Moreover, when examined by the colorimetric MTT assay[56], the PCL NPs presented minimal cytotoxicity against cells even upon UV-light (365 nm) illumination for DBTEO photocyclization (Supplementary Fig. 13). According to the minimal cytotoxicity, the PCL NPs hold potential for advanced cell imaging applications based on color-specific photoswitching.

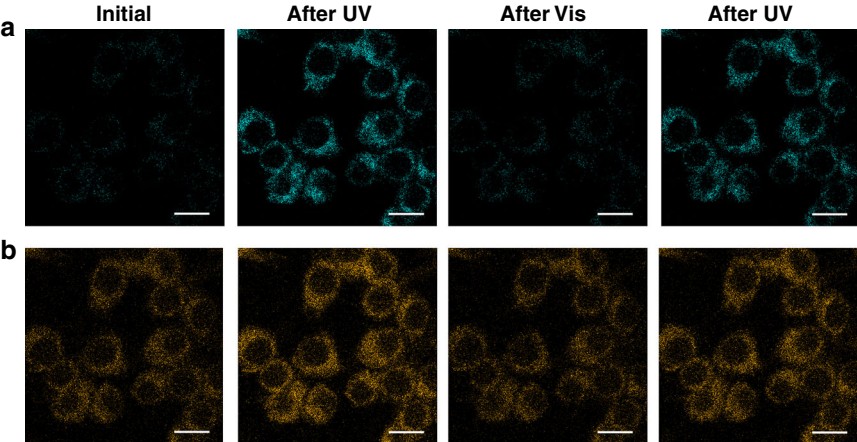

**Fig. 7** Bioimaging. Confocal microscopy images (scale bar: 10 μm) of the RAW264.7 cells incubated with the PCL nanoparticles for 1 h: (**a**) Blue emission channel (460–500 nm), and (**b**) orange emission channel (600–780 nm)

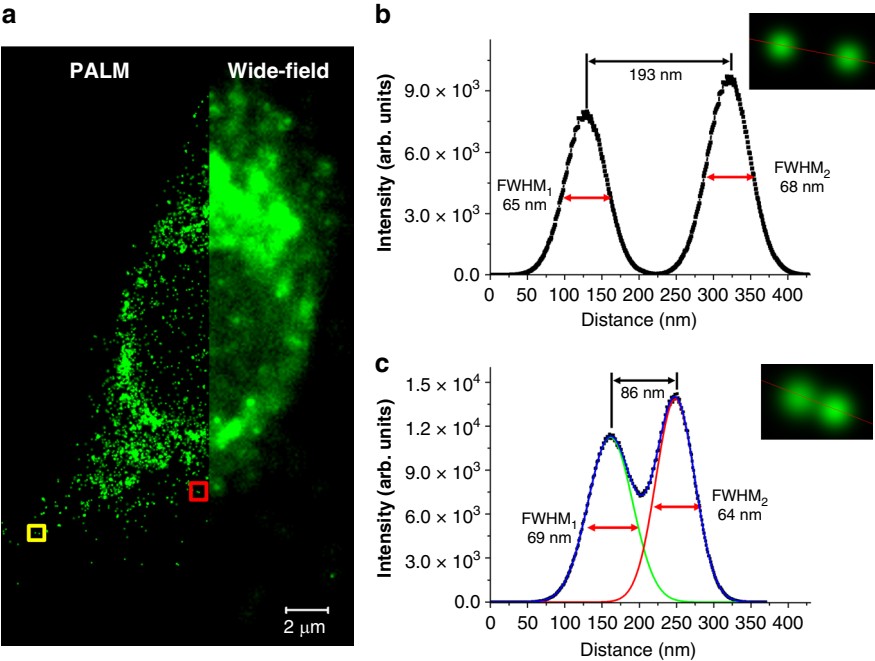

**Fig. 8** Super-resolution microscopic imaging. **a** Super-resolved PALM (left) and conventional wide-field (right) images of the RAW264.7 cell which is magnified a thousand times. **b**, **c** Fluorescence intensity cross-sectional profiles of two pairs of neighboring PCL nanoparticles in the square region in (**a**) (yellow square for (**b**) and red square for (**c**), respectively)

Finally, a super-resolution microscopy of a single RAW 264.7 cell was achieved by perfectly color-specific photoswitching under the PALM setup. As shown in Fig. 8a, it was found that the NPs are non-specifically endocytosed. In the region of the wide-field image, the individual NPs cannot be clearly recognized due to the close distance between them. In sharp contrast, the PALM images show clear nano-sized localizations of separated NPs. Figure 8b and c show the full width at half-maximum (FWHM) values of the two localized fluorescence points, which are very close within the diffraction limit (~200 nm). Each of the points are determined by Gaussian fitting to be as low as 64 nm, whose size is comparable with the hydrodynamic size of the NPs. It is noteworthy that the two local points as close as 86 nm could be well resolved (Fig. 8c). In addition, the overall spatial resolution was analyzed by using the Fourier ring correlation (FRC), which is a method recently often used to determine the resolution of localization-based super-resolution fluorescence imaging[57]. The

resolution of the localizations was determined to be about 73 nm (Supplementary Fig. 14), which is well matched with the above-mentioned FWHM values. These results indicate that the color-specific photoswitchable NPs are capable of cell imaging with a high resolution of about 70 nm that obviously goes beyond the diffraction limit of light.

## Discussion

In summary, we have developed a two-component dual-color fluorescence system showing perfect color-specific photoswitching. This system is composed of a blue fluorescent turn-on type diarylethene DBTEO and an orange color-emitting ESIPT fluorophore HPNIC. In our design, introducing a turn-on-type photochromic dye is a key to make the efficiency of fluorescence switching 100%. Furthermore, it is found out that highly frustrated energy transfer between DBTEO and HPNIC based on the

ESIPT process is prerequisite to achieve the color-specific photoswitching. We fabricated dual-emissive PCL polymer nanoparticles containing DBTEO and HPNIC with a hydrodynamic size of 57 nm. It was successfully demonstrated that the blue emission was selectively and reversibly turned on and off by photoswitching, while the orange emission visualized the internalization of nanoparticles into RAW264.7 cells independently and invariably. More importantly, based on the photoswitching capability of blue emission, we could obtain a super-resolution PALM image of a single cell with a resolution as low as 70 nm.

## Methods

**Materials**. All reagents and solvents were purchased from chemical suppliers (Sigma-Aldrich, Alfa Aesar Co., and TCI Co.), and used as received. All things necessary for reactions such as glassware and magnetic stirrer bars were cleaned and dried in an oven. Reactions were monitored through thin layer chromatography (TLC) plates (silica gel 60 F254, Merck Co.) upon UV-light (254 nm) illumination. Column chromatography was conducted with silica gel 60 (particle size 0.063–0.200 mm, Merck Co.).

**Measurements**. $^1$H NMR spectra were obtained by using a Bruker Avance 300 spectrometer (300 MHz). Mass (MS) spectra were performed on a JEOL JMS-600W/JMS-700GC, and elemental analyses (EA) were performed on a Flash EA 1112 (Thermo Fisher Scientific) instrument. Absorption and emission spectra were recorded on a Shimadzu UV-1650PC spectrophotometer and a Varian Cary Eclipse fluorescence spectrophotometer, respectively. Fluorescence lifetimes were measured by the time-correlated single-photon counting (TCSPC) technique, in which a FluoTime 200 instrument (Picoquant) including a 377-nm pulsed diode laser (fwhm ~70 ps) as an excitation light was used. The decay profiles of fluorescence were investigated by exponential fitting models through FluoFit Pro software. Absolute PL quantum yields were measured by employing a 3.2-in. integrating sphere in a Photon Technology International QM-40 spectrometer, and relative PL quantum yields were obtained using Fluorescein as a reference. The photographs of fluorescence images under illumination at 365 nm were acquired by a Canon (PowerShot G6) camera. A UV lamp (365 nm, 1.2 mW cm$^{-2}$) and a xenon lamp (300 W) equipped with a monochromator (Newport) or a color filter (> 420 nm of Newport) were used as a UV-light source and a visible-light source, respectively. Energy-Filtering Transmission Electron Microscope (EF-TEM) was performed with LIBRA 120 (Carl Zeiss).

**Calculation**. All DFT calculations were carried out in the gas phase using the Gaussian 09 quantum-chemical package, employing the B3LYP functionals with the 6–31 G(d,p) basis set[58]. The geometry optimizations for the ground state of molecules were performed, and vibration frequency calculations at the same level were performed for the obtained structures to confirm the stable minima[59]. The resulting geometries were used for time-dependent DFT (TD-DFT) calculations to describe the absorption processes of DBTEO and HPNIC. For calculating the emission process of closed-form DBTEO, the geometry optimization at the excited state of the molecule was applied using TD-DFT. On the other hand, the emission energy of keto-form HPNIC could be calculated by the geometry optimization process for the excited state of enol form HPNIC, in which the enol excited state was immediately converted to the keto excited state by "barrier-less" ESIPT process.

**Cell preparation for imaging**. RAW264.7 cell lines were provided by Korea Basic Science Institute and cultivated in the DMEM (Dulbecco's Modified Eagle's Medium) with 10% FBS, L-glutamine (5 × 10$^{-3}$ M), and gentamicin (5 μg mL$^{-1}$), in a humidified 5% CO$_2$ incubator at 37 °C. Prior to the experiment, 1 × 10$^5$ cells were seeded onto 35-mm coverglass bottom dishes and allowed to grow until a confluence of 60–70%. The cells were then washed twice with fresh PBS (pH 7.4) to remove the remnant growth medium and incubated in a fresh medium (2 mL) containing a photoswitchable PCL nanoparticle dispersion (DBTEO, 0.3 μg mL$^{-1}$; HPNIC, 0.2 μg mL$^{-1}$) for 1 h.

**Confocal fluorescence imaging**. A confocal microscope (TCS SP8 X, Leica) was used with a HC PL APO CS2 100x/1.4 oil-immersion objective. To switch on the blue emission of DBTEO, a 365-nm light from a UV lamp was illuminated for 2 min at a distance of 5 cm from the sample. To switch off the emission, a 458-nm argon laser was irradiated with 100% intensity for 5 min. A 405-nm excitation light was used for fluorescence imaging. The fluorescence detection ranges were from 460 nm to 500 nm for blue emission and from 600 nm to 780 nm for orange emission, respectively.

**Super-resolution fluorescence imaging**. Photoactivated localization microscopy (PALM) was conducted on an ELYRA P.1 microscope (Carl Zeiss) equipped with an oil-immersion objective (×100, 1.46 NA). The fluorescence images (100 nm per pixel) were acquired by using light sources (405 nm and 488 nm), a band pass filter (495–590 nm), and a detection camera (EM-CCD at 295 K). To obtain wide-field and super-resolved images, total 10,000 frames of the images were collected and analyzed with Zeiss Zen software. The wide-field image was reconstructed from the default value of the software. The super-resolved image was reconstructed by analyzing all frames, in which fluorescent spots between 70 and 200 nm were specifically selected to discard the unreasonable patterns. The super-resolved image (10 nm per pixel) was rendered by selecting the fluorescent spots, whose localization precision was from 10 to 50 nm, to exclude the poorly localized fluorescent probes. ImageJ plugin was used to analyze the Fourier ring correlation (FRC) of the localizations.

**Preparation and structural analytical data of DBTEO**. DBTEO was synthesized by modified synthetic literature[43]. 1,2-bis(2-ethyl-1-benzothiophen-3-yl)perfluorocyclopentene[60] (3.0 g, 6.04 mmol) was dissolved in acetic acid (120 mL), and the mixture was heated to 118 °C. In all, 25 mL of 35% hydrogen peroxide solution was slowly injected to the solution, and the mixture was stirred and refluxed for 30 min. The resulting mixture was poured into a huge amount of water to give a precipitate. The precipitate was filtered, washed with H$_2$O, and concentrated. Reprecipitation in methanol from dichloromethane solution afforded white powder (2.17 g, yield = 64%). Analytical data of DBTEO: $^1$H NMR (300 MHz, CDCl3, δ): 1.05 (t, $J$ = 7.6 Hz, 3.7 H (ap)), 1.40 (t, $J$ = 7.6 Hz, 2.3 H (p)), 2.32–2.48 (m, 1.6 H (p)), 2.51–2.70 (m, 2.4 H (ap)), 7.14 (d, $J$ = 6.9 Hz, 0.8 H (p)), 7.20 (d, $J$ = 7.3 Hz, 1.2 H (ap)), 7.39–7.48 (m, 1.6 H (p)), 7.54–7.69 (m, 2.4 H (ap)), 7.68 (d, $J$ = 6.8 Hz, 0.8 H (p)), 7.76 (d, $J$ = 7.2 Hz, 1.2 H (ap)). HRMS (FAB +, m/z): [M + H]$^+$ calculated for C$_{25}$H$_{19}$F$_6$O$_4$S$_2$, 561.0629; found, 561.0630. Elemental analysis, calculated for C$_{25}$H$_{18}$F$_6$O$_4$S$_2$: C, 53.57; H, 3.24; S, 11.44; O, 11.42. Found: C, 53.60; H, 3.19; S, 11.43; O, 11.42.

**Preparation and structural analytical data of HPNIC**. HPNIC was synthesized according to the literature[61]. Analytical data of HPNIC: $^1$H NMR (300 MHz, CDCl3, δ): 7.14–7.20 (m, 4 H), 7.28 (d, 1 H), 7.36 (m, 1 H), 7.44 (s, 1 H), 7.54 (t, 1 H), 7.62–7.72 (m, 4 H), 7.75–7.87 (m, 4 H), 8.70–8.79 (m, 3 H), 13.50 (s, 1 H). $^{13}$C NMR (125 MHz, CDCl3, δ): 111.85, 115.54, 121.27, 122.80, 122.86, 123.36, 123.46, 124.46, 125.72, 126.03, 126.08, 126.42, 126.82, 126.92, 127.59, 127.62, 127.81, 128.63, 128.75, 129.42, 129.92, 130.90, 131.22, 135.04, 135.17, 139.42, 148.10, 155.89. MS m/z: 436 (M +). Elemental analysis, calculated for C$_{31}$H$_{20}$N$_2$O: C, 85.30; H, 4.62; N, 6.42; O, 3.67. Found: C, 85.51; H, 4.75; N, 6.58; O, 3.70.

**Preparation of PCL nanoparticle containing DBTEO and HPNIC**. PCL nanoparticle was fabricated by modified literature procedures[52]. A solution of PCL (25 mg, M$_n$ ~10,000, Aldrich), DBTEO (0.15 mg, 0.6 wt% to PCL), and HPNIC (0.10 mg, 0.4 wt% to PCL) in toluene (1 mL) was stirred for 1 h at room temperature to be dissolved clearly. A portion of the mixture toluene solution (20 μL) was injected into an aqueous micellar solution containing Pluronic F-68 surfactant (Aldrich) (50 mg of F-68 was dissolved in 5 mL of deionized water). Then, the resulting aqueous solution was emulsified with sonication for 1 min (pulse: on-time 2 s and off-time 3 s) by using an ultrasonic probe of the titanium alloy (power: 60 W). The emulsified oil-in-water solution was heated by a heat gun and simultaneously evaporated via rotary evaporator to remove toluene completely. To this solution, another portion of the mixture toluene solution (20 μL) was injected, and the whole solution was re-emulsified with sonication and evaporated in the same way as the above. In this manner, the procedure of emulsification and evaporation was repeated five times in total. The final amounts in the solution are 2.5 mg of PCL, 50 mg of F-68, 0.015 mg of DBTEO, and 0.010 mg of HPNIC in 5 mL of water.

## Data availability

The data supporting the findings of this study are available within the article and its Supplementary Information or from the corresponding author upon reasonable request.

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

## Acknowledgements

This work was supported by the Creative Research Initiative Program through the National Research Foundation of Korea (NRF) funded by the Ministry of Science, ICT &

Future Planning (MSIP) (Grant No. 2009-0081571[RIAM0417-20170011]), and the Basic Science Research Program through the NRF funded by the MSIP (Grant No. 2017R1E1A1A01075372). S.K. thanks Korea Research Institute of Standards and Science (KRISS-2018-GP2018-0018) for KIST intramural program. We thank Jinhoe Hur and Hongchan Joung (Optical Biomed Imaging Center, Ulsan National Institute of Science and Technology (UNIST), Korea) for technical support and discussions on PALM imaging and analysis.

## Author contributions

D.K. designed the research. S.Y.P. supervised the research. S.K. and S.Y.P. offered intellectual input. D.K. synthesized molecules and performed spectroscopic experiments. D.K. and J.E.K. performed computational calculations. D.K., K.J. and H.P. fabricated nanoparticles. K.J., H.P. and S.L. prepared cell experiments and performed fluorescence imaging. D.K. performed super-resolution imaging. D.K. and S.Y.P. wrote the first draft of the paper. D.K., K.J., J.E.K., S.K. and S.Y.P. contributed to the final version of the paper.

## Additional information

**Competing interests:** The authors declare no competing interests.

**Peer Review Information**: *Nature Communications* thanks Daniel Gryko and other anonymous reviewers for their contribution to the peer review of this work. Peer reviewer reports are available.

