## [Peer Review File · Nature Communications]

Reviewers' comments:

Reviewer #1 (Remarks to the Author):

The manuscript by Park and co-workers is an excellent piece of work. It is focused on nanoparticles displaying dual fluorescence. This fluorescence can be modulated by switching-off one component of the system. Photoswitchable nanoparticles have been object of study for some time already, but this manuscript reveals a completely novel strategy i.e. complete switch off of emission of one component by alternating UV irradiation and light excitation. Importantly, this system comprised of two different chromophores reaches 100% switching on-off, which has never been achieved before. This type of discovery is not of only academic importance but it also has the clear use in fluorescence imaging. Authors have also presented preliminary results with cells using PALM super-resolution microscopy technique. The manuscript has been prepared by one of leaders in this field and as such it does not have any major deficiencies. All photophysical studies, materials studies and computational studies have been performed according to state-of-the-art. The concept is elegant but at the same time it is rather simple to follow and it is quite possible that other research groups will 'join'. The novelty of this manuscript is beyond the shadow of the doubts.

Minor points:

1. The details of the preparation of nanoparticles have to be added.
2. The success of the concept relies on two group of compounds: photoswitchable diarylethylenes and 2-(2-hydroxyphenyl)imidazoles. Yet neither key papers on photoswitchable diarylethylenes by Kellog and Irie were cited, nor papers on ESIPT-capable 2-(2-hydroxyphenyl)imidazoles and their pi-expanded analogs were mentioned (except for author's own papers). I strongly suggest to add some citations her and there.

Reviewer #2 (Remarks to the Author):

The work describes a photochromic system whereby the short wavelength emitter is photswitched on and off without compromising the long wavelength emitter. This is unusual, as under normal conditions FRET should occur. The goal is achieved by combining a fluorophore comprising an extraordinarily large Stokes shift due to a enol – keto transition, with a fluorophore where an internal photoswitched ring closure / opening reaction tunes the absorption spectrum by 100 nm. As a possible application, the authors show superresolutino microscopy using the PALM format, which itself is currently a very important and timely field of research.

In my opinion, this is a very interesting system of fluorophores with applications beyond the specific example of PALM microscopy. It should as well be applicable in a RESOLFT type of nanoscopy as well as (possibly in combination with more fluorophores) in many sophisticated regimes of FRET in order to entangle, for instance, protein interacitons in cell ohysiological studies. With this broad range of application, it deserves publication in a well visible journal.

The paper is well written and contains all necessary information. I just suggest some minor issues prior to publication:

1. References: Credit should be given to those scientists who published things first rather than (only) referencing secondary and tertiary work. Superresolution by optically induced switching of chromophores has been postulated by Wichmann, Hell in 1994 (Opt. Lett. 19, 11) and was experimentally shown in 1999 (Opt. Lett 24: 954). A switchable chromophore in a FRET type setting has first been used for diffraction unlimited resolution in 2006 (Angew. Chem Int. Ed. 45, 7462) (New J. Phys. 8, 275).
2. Use the terms "open" and "closed" forms only for DBTEO. Do not use the term „photoswitched“ because it is unclear, which form is meant by „photoswitched“, as both transitions O->C and C->O are actually triggered by photoswitching.

Reviewer #3 (Remarks to the Author):

In this manuscript, Park and co-workers have developed a dual-color fluorescent nanoparticle based on photochromic diarylethenes for bio-imaging and super-resolution microscopy. Photochromic diarylethenes were used as a switchable fluorophore as the closed isomer exhibited strong emission while the open isomer was non-emissive. The ESIPT-based HPNIC was used as another fluorophore, whose emission had no interaction with the photoswitching diarylethenes. The authors claimed that the non-variable emission from HPNIC could be used as a tracking fluorescence while the emission from the diarylethene was used to perform a switchable signal for bio-imaging as well as super-resolution microscopy. This work is interesting, while it is just an extensive application of the author's previous work (*Adv. Opt. Mater.* 2016, 4, 790; ref 40) and we don't see any great breakthrough that could qualify this paper to publish on *Nature Communications*.

Major comments:

1. The authors stated in the title for photoswitchable bio-imaging and super-resolution microscopy with dual-color nanoparticles. Yet, in major part of the manuscript, the authors just discussed the fluorescence photoswitching of diarylethene and non-FRET process between diarylethene and HPNIC in PMMA film. Although the authors showed that the fluorescence performances were similar in PCL film with PMMA film, the absorption behaviors were different in PCL nanoparticles, as shown in Figure S6a. We suggest the photochemical properties of this bi-molecular system should be studied mainly in nanoparticles, where the bio-imaging and super-resolution microscopy were conducted. Moreover, the fluorescence and photochromic quantum yields of diarylethene and HPNIC should be provided in PCL nanoparticles, respectively.
2. The emission wavelength of diarylethene is not suitable for bio-imaging as blue fluorescence can be easily interfered by cell background. In fact, Irie et al has developed fluorescent diarylethenes with orange emission as well as high quantum yield with proper modification of the diarylethene core, and red/near-infrared fluorescent dyes can be utilized as the tracking fluorophore or inner reference for this dual-color bio-imaging.
3. Dual-color fluorescence has its advantage, yet, the always-on of the orange emission would also cause significant background signals.

Other comments:

1. Page 9, line 212-214. The authors stated that biocompatibility as well as structural similarity of PCL to PMMA, it is anticipated to provide similar surrounding environments to the fluorophores. Why similar biocompatibility should give similar surrounding environments like polarity? And references should be provided on the similar biocompatibility between PCL and PMMA.
2. Page 9, line 222-223. The authors stated that the nanoparticle is tiny enough for facile delivery into the cells. References should be provided to prove this point.
3. Besides the MTT for cytotoxicity, the phototoxicity should also be examined, as UV light was used during the cell imaging.

Summary of Revisions

We are very grateful to the reviewers for important and helpful comments on our manuscript. We have thoroughly revised our manuscript based on the comments. Our point-by-point responses to the reviewer's comments and the corresponding revisions made in the revised manuscript are described below.

Reviewer #1 (Remarks to the Author):

The manuscript by Park and co-workers is an excellent piece of work. It is focused on nanoparticles displaying dual fluorescence. This fluorescence can be modulated by switching-off one component of the system. Photoswitchable nanoparticles have been object of study for some time already, but this manuscript reveals a completely novel strategy i.e. complete switch off of emission of one component by alternating UV irradiation and light excitation. Importantly, this system comprised of two different chromophores reaches 100% switching on-off, which has never been achieved before. This type of discovery is not of only academic importance but it also has the clear use in fluorescence imaging. Authors have also presented preliminary results with cells using PALM super-resolution microscopy technique. The manuscript has been prepared by one of leaders in this field and as such it does not have any major deficiencies. All photophysical studies, materials studies and computational studies have been performed according to state-of-the-art. The concept is elegant but at the same time it is rather simple to follow and it is quite possible that other research groups will 'join'. The novelty of this manuscript is beyond the shadow of the doubts.

Response and Revision: We are very grateful to the reviewer for his/her positive evaluation and compliments.

Minor points:

1. The details of the preparation of nanoparticles have to be added.

Response and Revision: According to the reviewer's suggestion, we have added further details about the preparation of nanoparticles as below.

In the revised manuscript (p. 15, line 13),

“Preparation of PCL nanoparticle containing DBTEO and HPNIC: PCL nanoparticle was fabricated by modified literature procedures.⁵² A solution of PCL (25 mg, $M_n \sim 10,000$, Aldrich), DBTEO (0.15 mg, 0.6 wt% to PCL), and HPNIC (0.10 mg, 0.4 wt% to PCL) in toluene (1 mL) was stirred for 1 h at room temperature to be dissolved clearly. A portion of the mixture toluene solution (20 μ L) was injected into an aqueous micellar solution containing Pluronic F-68 surfactant (Aldrich)

(50 mg of F-68 was dissolved in 5 mL deionized water). Then, the resulting aqueous solution was emulsified with sonication for 1 min (pulse: on-time 2 sec and off-time 3 sec) by using an ultrasonic probe of the titanium alloy (power: 60 W). The emulsified oil-in-water solution was heated by a heat gun and simultaneously evaporated via rotary evaporator to remove toluene completely. To this solution, another portion of the mixture toluene solution (20 μ L) was injected, and the whole solution was re-emulsified with sonication and evaporated in the same way as the above. In this manner, the procedure of emulsification and evaporation was repeated five times in total. The final amounts in the solution are 2.5 mg of PCL, 50 mg of F-68, 0.015 mg of DBTEO, and 0.010 mg of HPNIC in 5 mL water.”

2. The success of the concept relies on two group of compounds: photoswitchable diarylethylenes and 2-(2-hydroxyphenyl)imidazoles. Yet neither key papers on photoswitchable diarylethylenes by Kellogg and Irie were cited, nor papers on ESIPT-capable 2-(2-hydroxyphenyl)imidazoles and their pi-expanded analogs were mentioned (except for author’s own papers). I strongly suggest to add some citations here and there.

Response and Revision: We appreciate this important and helpful comment. As the reviewer recommended, we have cited more literatures (*J. Org. Chem.* **1967**, 32, 3093; *J. Org. Chem.* **1988**, 53, 803), which are key papers for developing diarylethene reported by Kellogg and Irie, respectively. A key paper on imidazole-based ESIPT molecules (Mosquera, M. et al., *J. Phys. Chem.* **1996**, 100, 5398) has also been cited in the revised manuscript.

Added references:

43. Kellogg, R. M., Groen, M. B. & Wynberg, H. Photochemically induced cyclization of some furyl- and thienylethenes. *J. Org. Chem.* **32**, 3093-3100 (1967).

44. Irie, M. & Mohri, M. Thermally irreversible photochromic systems. Reversible photocyclization of diarylethene derivatives. *J. Org. Chem.* **53**, 803-808 (1988).

51. Mosquera, M., Penedo, J. C., Ríos Rodríguez, M. C. & Rodríguez-Prieto, F. Photoinduced Inter- and Intramolecular Proton Transfer in Aqueous and Ethanolic Solutions of 2-(2'-Hydroxyphenyl)benzimidazole: Evidence for Tautomeric and Conformational Equilibria in the Ground State. *J. Phys. Chem.* **100**, 5398-5407 (1996).

Reviewer #2 (Remarks to the Author):

The work describes a photochromic system whereby the short wavelength emitter is photswitched on and off without compromising the long wavelength emitter. This is unusual, as under normal

conditions FRET should occur. The goal is achieved by combining a fluorophore comprising an extraordinarily large Stokes shift due to an enol – keto transition, with a fluorophore where an internal photoswitched ring closure / opening reaction tunes the absorption spectrum by 100 nm. As a possible application, the authors show super-resolution microscopy using the PALM format, which itself is currently a very important and timely field of research.

In my opinion, this is a very interesting system of fluorophores with applications beyond the specific example of PALM microscopy. It should as well be applicable in a RESOLFT type of nanoscopy as well as (possibly in combination with more fluorophores) in many sophisticated regimes of FRET in order to entangle, for instance, protein interactions in cell physiological studies. With this broad range of application, it deserves publication in a well visible journal.

The paper is well written and contains all necessary information. I just suggest some minor issues prior to publication:

1. References: Credit should be given to those scientists who published things first rather than (only) referencing secondary and tertiary work. Superresolution by optically induced switching of chromophores has been postulated by Wichmann, Hell in 1994 (*Opt. Lett.* 19, 11) and was experimentally shown in 1999 (*Opt. Lett.* 24: 954). A switchable chromophore in a FRET type setting has first been used for diffraction unlimited resolution in 2006 (*Angew. Chem Int. Ed.* 45, 7462) (*New J. Phys.* 8, 275).

Response and Revision: We are very grateful to the reviewer for the positive evaluation and helpful comments/suggestions. According to the reviewer's recommendation, we have cited the reference literatures (*Opt. Lett.* **1994**, 19, 11, 780; *Opt. Lett.* **1999**, 24, 14, 954; *Angew. Chem. Int. Ed.* **2006**, 45, 7462; *New J. Phys.* **2006**, 8, 275), which are the original papers reporting super-resolution imaging concept, experimental super-resolution imaging, and photoswitchable FRET systems for super-resolution imaging, respectively.

Added references:

17. Hell, S.W. & Wichmann, J. Breaking the diffraction resolution limit by stimulated emission: stimulated-emission-depletion fluorescence microscopy. *Opt. Lett.* **19**, 780-782 (1994).

18. Klar, T. A. & Hell, S. W. Subdiffraction resolution in far-field fluorescence microscopy. *Opt. Lett.* **24**, 954-956 (1999).

19. Bossi, M., Belov, V., Polyakova, S. & Hell, S. W. Reversible red fluorescent molecular switches. *Angew. Chem. Int. Ed.* **45**, 7462-7465 (2006).

20. Bossi, M., Fölling, J., Dyba, M., Westphal, V. & Hell, S. W. Breaking the diffraction

resolution barrier in far-field microscopy by molecular optical bistability. *New J. Phys.* **8**, 275 (2006).

2. Use the terms “open” and “closed” forms only for DBTEO. Do not use the term „photoswitched“ because it is unclear, which form is meant by „photoswitched“, as both transitions O->C and C->O are actually triggered by photoswitching.

Response and Revision: We appreciate this important comment. As the reviewer suggested, we have changed “photochromic/photoswitched” to “photocyclization/photocycloreversion”. We have also added the expressions of open form (**O** form) and closed form (**C** form) throughout the entire manuscript to make it clearer for readers.

In the manuscript:

“the **photocyclization** reaction occurred to produce the **C** form” (p.6, line 5)

“the **cycloreversion** reaction from **C** form to **O** form of DBTEO” (p.7, line 5)

“the lifetime of DBTEO (**C** form)” (p.7, line 10)

“the photochromic conversion between **O** form and **C** form of DBTEO” (p.7, line 14)

“the photochromic reaction between **O** form and **C** form of DBTEO” (p.7, line 26)

“from DBTEO (**C** form) to Nile Red” (p.8, line 4)

“the blue emission of DBTEO (**C** form)” (p.10, line 16)

“the blue emission of DBTEO in the **C** form” (p.10, line 18)

Reviewer #3 (Remarks to the Author):

In this manuscript, Park and co-workers have developed a dual-color fluorescent nanoparticle based on photochromic diarylethenes for bio-imaging and super-resolution microscopy. Photochromic diarylethenes were used as a switchable fluorophore as the closed isomer exhibited strong emission while the open isomer was non-emissive. The ESIPT-based HPNIC was used as another fluorophore, whose emission had no interaction with the photoswitching diarylethenes. The authors claimed that the non-variable emission from HPNIC could be used as a tracking fluorescence while the emission from the diarylethene was used to perform a switchable signal for bio-imaging as well as super-resolution microscopy. This work is interesting, while it is just an extensive application of the author’s previous work (*Adv. Opt. Mater.* 2016, 4, 790; ref 40) and we don’t see any great breakthrough that could qualify this paper to publish on Nature Communications.

Response and Revision: We appreciate many helpful and important comments of Reviewer #3

including the three major comments; (1) measuring PLQY and photochromic QY in PCL nanoparticle, (2) blue emission of diarylethenes for bioimaging, (3) concern about the orange emission generating significant background signal. In the revised manuscript, we have fully addressed these issues as well as the three minor comments by making supplementary experiments and adding detailed explanations (*vide infra*). We are indebted to the Reviewer #3 for such remarkable improvement of our manuscript based on his/her comments. We regret however our original manuscript could not convince Reviewer #3 of the breakthrough novelty and significance included in it. Unfortunately, Reviewer #3 considered that our present work is a simple extension of our previous work (*Adv. Opt. Mater.* **2016**, 4, 790), which is not true in terms of the totally different switching principles and materials system employed for the present and past works. While the previous work is a three component system comprising two ESIPT dyes and one diarylethene switch, the present work is a two component system comprising fluorescent photochromic diarylethene and one ESIPT dye. Not only the number of components, these two systems are totally different in energy transfer schemes enabling only the present system to demonstrate 100% perfect color-specific photoswitching for the first time. This aspect was in fact highly appreciated and complimented by both Reviewers #1 and #2 as a completely novel strategy. Herein, we solicit kind reconsideration of Reviewer #3 of our revised manuscript in terms of the novelty issue. To help his/her reconsideration, we summarize the state-of-the-art research trends as follows. Although a few color-specific photoswitching systems, which have typically employed three components (*i.e.*, two fluorophores and one photochromic switch), have been reported by several research groups including us (*e.g. Nano Lett.* **2012**, 12, 3537; *J. Am. Chem. Soc.* **2012**, 134, 12091; *J. Am. Chem. Soc.* **2013**, 135, 3208; *Chem. Mater.* **2013**, 25, 2495; *Adv. Opt. Mater.* **2016**, 4, 790), the switching efficiency has been limited only up to 50 ~ 80% because delicate manipulation of all the energy transfer crosstalks among the three components in the systems is unfeasible. In this regard, we believe that our present study of two-component system consisting of an ESIPT dye and a turn-on diarylethene represents a breakthrough to make the switching efficiency “perfect (100%)” in the color-specific photoswitching. Also, in a practical point of view, our two-component system is a lot simpler than the conventional three-component ones, thus it can be more conveniently and reproducibly prepared and applied into many practical applications. As a proof of concept, we applied our perfect color-specific photoswitching system in a form of biocompatible nanoparticles to bioimaging and super-resolution microscopy. To the best of our knowledge, the demonstration of both bioimaging and super-resolution microscopy by using color-specific photoswitching system has been achieved for the first time in this study although super-resolution RESOLFT imaging using color-specific switching was demonstrated by M. L. Bossi et al. (*Small* **2008**, 4, 134) and bioimaging using color-specific switching was demonstrated by several groups (*Chem. Eur. J.* **2012**, 18, 3122; *RSC Adv.* **2014**, 4, 15613; *Nanoscale* **2015**, 7, 11263).

Major comments:

1. The authors stated in the title for photoswitchable bio-imaging and super-resolution microscopy with dual-color nanoparticles. Yet, in major part of the manuscript, the authors just discussed the fluorescence photoswitching of diarylethene and non-FRET process between diarylethene and HPNIC in PMMA film. Although the authors showed that the fluorescence performances were similar in PCL film with PMMA film, the absorption behaviors were different in PCL nanoparticles, as shown in Figure S6a. We suggest the photochemical properties of this bi-molecular system should be studied mainly in nanoparticles, where the bio-imaging and super-resolution microscopy were conducted. Moreover, the fluorescence and photochromic quantum yields of diarylethene and HPNIC should be provided in PCL nanoparticles, respectively.

Response and Revision: We appreciate this critical and helpful comment of the reviewer. According to the reviewer's recommendation, we have studied the photophysical properties including absorption, fluorescence, and fluorescence quantum yields of DBTEO and HPNIC in PCL nanoparticles in more detail. As the reviewer pointed out, there are scattering effects in the absorption spectra in Figure S6a (= Figure S8a in the revised manuscript). To exactly obtain the absorption and emission spectra of the DBTEO and HPNIC by reducing the scattering effects, the concentrations of the dyes in the PCL nanoparticles were increased to 4 wt% with respect to the PCL polymer matrix. As shown in Figure S6 and S7 (also in Table S1) in the revised manuscript, the absorption and emission properties (*i.e.* spectral shapes and maximum wavelengths) of the dyes in PCL nanoparticles are virtually identical to those in PMMA and PCL films (see Figure S5 in the Supporting Information). Also, we have measured the relative photoluminescence quantum yield of the two dyes in PCL nanoparticle (0.48 for DBTEO and 0.12 for HPNIC, respectively) by using Fluorescein as a reference, well corresponding with the values in PMMA and PCL films. This result is attributed to the fact that the surrounding environment for dyes are identical in both the PCL film and nanoparticles. In the revised manuscript, the absorption spectra, emission spectra, and photoluminescence quantum yields have been added as Figure S6,7 and Table S1, and the following explanation has also been added.

“Notably, these optical properties including absorption, emission, and fluorescence quantum yield were negligibly changed in the PCL NPs except for some scattering effect in absorption spectra (see Figure S6,7 and Table S1).” (p.9, line 9)

Figure S6. (a) UV-vis absorption spectra of the PCL nanoparticles doped with DBTEO (4 wt% to PCL matrix) upon UV and visible light irradiations. (b) Differential between the absorptions of the initial state and photostationary state (PSS). Two blue wavelengths, 312 nm and 415 nm, are assigned to the absorption maximum wavelengths of open and closed form DBTEO, respectively. (c) Photoluminescence (PL) spectra of DBTEO in PCL nanoparticle (4 wt% to PCL matrix) upon UV and visible light irradiations.

Figure S7. (a) UV-vis absorption spectrum and (b) photoluminescence (PL) spectrum of the PCL nanoparticles doped with HPNIC (4 wt% to PCL matrix).

Table S1. Absorption wavelength, fluorescence wavelength, and fluorescence quantum yield of DBTEO and HPNIC in PMMA films, PCL films, and PCL nanoparticles.

	Absorption maximum ($\lambda_{\text{abs,max}}/\text{nm}$)		Fluorescence maximum ($\lambda_{\text{emi,max}}/\text{nm}$)		Fluorescence quantum yield (Φ_F)	
	DBTEO	HPNIC	DBTEO	HPNIC	DBTEO	HPNIC
PMMA film	312(o), 415(c)	337	500	590	0.55 ^a	0.11 ^a
PCL film	312(o), 415(c)	337	504	594	0.50 ^a	0.11 ^a
PCL nanoparticle	312(o), 415(c)	337	504	594	0.48 ^b	0.12 ^b

^aAbsolute PL quantum yields were obtained using a QM-40 spectrophotometer equipped with an integrating sphere. ^bRelative PL quantum yields were obtained using fluorescein as a reference dye.

In addition, we have also measured the photochromic quantum yields of DBTEO in PCL nanoparticles. To evaluate them, the change of absorbance at 415 nm (a characteristic absorption peak of closed-ring DBTEO), was recorded upon light irradiations (at 365 nm for photocyclization and 415 nm for photocycloreversion) as shown in Figure S10 and S11. As the slope in the figures is proportional to the photochromic quantum yield, the photocyclization and photocycloreversion quantum yields of DBTEO in PCL nanoparticles can be calculated to be 0.20 and 0.083, respectively, with respect to a reference system ($\Phi_{\text{o} \rightarrow \text{c, DBTEO}} = 0.28$, $\Phi_{\text{c} \rightarrow \text{o, DBTEO}} = 0.18$ in dioxane) (*J. Am. Chem. Soc.* **2011**, 133, 13558). Both the photochromic quantum yields in the PCL nanoparticle system were lower than those in dioxane solution system, which might be due to the rigid surrounding of the PCL matrix. In the revised manuscript, the absorption changes upon light irradiations have been added as Figure S10 and S11, and following explanation has also been added.

“The photocyclization and photocycloreversion quantum yields of DBTEO in PCL NPs were measured to be 0.20 and 0.083, respectively, in which DBTEO in dioxane solution was used as a reference (Figure S10 and S11).⁴⁹ Both photochromic quantum yields of DBTEO in PCL NPs are lower than those (0.28 and 0.18, respectively) in dioxane solution, which is most likely attributed to the rigid environment of the PCL matrix.” (p.10, line 5)

Figure S10. Photocyclization quantum yield measurement. (a) Corrected absorption of DBTEO at 415 nm in dioxane solution upon continuous irradiation of 365 nm light. (b) Corrected absorption of DBTEO at 415 nm in PCL nanoparticles upon continuous irradiation of 365 nm light. From the comparison of two slopes in (a) and (b), the photocyclization quantum yield of DBTEO in PCL nanoparticles is calculated from the reference value (0.28) in dioxane solution, resulting in a value of 0.20.

Figure S11. Photocycloreversion quantum yield measurement. (a) Corrected absorption of DBTEO at 415 nm in dioxane solution upon continuous irradiation of 415 nm light. (b) Corrected absorption of DBTEO at 415 nm in PCL nanoparticles upon continuous irradiation of 415 nm light. From the comparison of two slopes in (a) and (b), the photocycloreversion quantum yield of DBTEO in PCL nanoparticles is calculated from the reference value (0.18) in dioxane solution, resulting in a value of 0.083.

2. The emission wavelength of diarylethene is not suitable for bio-imaging as blue fluorescence can be

easily interfered by cell background. In fact, Irie et al has developed fluorescent diarylethenes with orange emission as well as high quantum yield with proper modification of the diarylethene core, and red/near-infrared fluorescent dyes can be utilized as the tracking fluorophore or inner reference for this dual-color bio-imaging.

Response and Revision: As the reviewer commented, red/near-infrared fluorescence has several advantages in bioimaging field such as low autofluorescence, deep penetration depth, and low phototoxicity. However, in this study, we focused on presenting the unprecedented design principle of perfect color-specific photoswitching system and demonstrating proof-of-concept.

Although blue/green emissive probes have several disadvantages for in vivo bioimaging, they are widely used and studied for in vitro cell imaging (*Eur. J. Physiol.* **2013**, 465, 347-359); for example, DAPI (4,6-diamidino-2-phenylindole) is a commercially available blue probe ($\lambda_{\text{max,exc}} = 358$ nm, $\lambda_{\text{max,emi}} = 461$ nm) that stains cell nuclei. As shown in Figure 7 in the manuscript, the bright blue emission of DBTEO was also clearly observed from cells, because the fluorescence quantum yield of DBTEO is high enough to distinguish its own fluorescence from the cell background fluorescence. Moreover, the fluorescence of DBTEO is switchable; that is, its on/off modulation is synchronized with the intentional photoswitching operation at different wavelengths, thus being easily distinguishable from the static emission of cell autofluorescence.

In these regards, we believe that blue-emitting DBTEO is suitable to demonstrate our concept in the present study. In the follow-up research of this study, we will follow the reviewer's suggestion to develop red/NIR-emitting diarylethenes to realize this concept under in vivo conditions.

3. Dual-color fluorescence has its advantage, yet, the always-on of the orange emission would also cause significant background signals.

Response and Revision: We appreciate this important comment. We could infer two issues from this reviewer's comment: (i) the orange emission can be interfered with background cell autofluorescence, (ii) the orange emission can act as a background signal for the read-out of the blue emission. For the former, as shown in Figure S12a, the orange HPNIC signal observed from cells after 1-3 h incubation with the PCL nanoparticles was clearly distinguished from the cell autofluorescence in the control image, revealing that the interference between the HPNIC signal and cell autofluorescence is negligible. To clarify this, HPNIC signals obtained from cell images has been newly added in Figure S12b in the revised manuscript.

Figure S12. (a) Fluorescence images (orange region) of RAW264.7 cells indicating the intracellular delivery of PCL nanoparticles containing DBTEO and HPNIC after 1-3 h incubation. (b) Cellular fluorescence intensity obtained from the fluorescence images (a).

For the latter, the interference between blue and orange emissions can be technically avoided by separating the read-out channels. As shown in Figure R1, when the read-out wavelength ranges for blue channel is set as 460-500 nm (see the yellow shaded region in the figure), the blue emission can be read-out without interference from the orange emission. In fact, we separated the two emission channels (460-500 nm for blue emission and 600-780 nm for orange emission, respectively) for confocal microscopy images in Figure 7 in the manuscript, and successfully obtained fluorescent images at each channel with no significant interference.

Figure R1. Normalized photoluminescence (PL) spectra of DBTEO and HPNIC in PMMA and PCL films. Yellow shaded region represents a wavelength width (460-500 nm) of blue emission channel.

Other comments:

1. Page 9, line 212-214. The authors stated that biocompatibility as well as structural similarity of PCL to PMMA, it is anticipated to provide similar surrounding environments to the fluorophores. Why similar biocompatibility should give similar surrounding environments like polarity? And references should be provided on the similar biocompatibility between PCL and PMMA.

Response and Revision: Thank you for this important comment. Actually, our original intention was that PCL was chosen as polymer nanoparticle matrix not only because it is approved as biocompatible polymer by the Food and Drug Administration (FDA) for use in bio-applications such as drug delivery, but also because it would provide the fluorophores with surrounding environments (*e.g.*, polarity) similar to that of PMMA due to the structural similarity. For clarity, the full sentence “With biocompatibility as well as structural similarity to PMMA, it is anticipated that PCL would provide the fluorophores with surrounding environments (*e.g.*, polarity) similar to that of PMMA.” has been revised and a relevant reference has been cited as below.

In the manuscript,

“We chose PCL as a polymeric container because it is approved as a biocompatible polymer by the Food and Drug Administration (FDA) for use in bio-applications such as drug delivery.⁵³ In addition, it is anticipated that PCL would provide the fluorophores with surrounding environments (*e.g.*, polarity) similar to that of PMMA due to the structural similarity.” (p.9, line 4)

Added reference:

53. Mondal, D., Griffith, M. & Venkatraman, S. S. Polycaprolactone-based biomaterials for tissue engineering and drug delivery: Current scenario and challenges. *Int. J. Polym. Mater. Polym. Biomater.* **65**, 255-265 (2016).

2. Page 9, line 222-223. The authors stated that the nanoparticle is tiny enough for facile delivery into the cells. References should be provided to prove this point.

Response and Revision: Thank you for this helpful comment. As the reviewer recommended, we have cited the reference literature (*Nature Nanotech.* **2008**, 3, 145-150), which demonstrated that tiny nanoparticles with the size range of 25-50 nm can be delivered into cells most efficiently.

Added reference:

54. Jiang, W., Kim, B. Y., Rutka, J. T. & Chan, W. C. Nanoparticle-mediated cellular response is size-dependent. *Nat. Nanotechnol.* **3**, 145-150 (2008).

3. Besides the MTT for cytotoxicity, the phototoxicity should also be examined, as UV light was used

during the cell imaging.

Response and Revision: We appreciate this important and helpful comment. According to the reviewer's suggestion, we have carried out the phototoxicity test with RAW 264.7 cells under the illumination of 365 nm light for 2 and 4 minutes, which was the experimental condition for confocal cell imaging. As can be seen in Figure S13b, the photo-induced cytotoxicity upon UV-light (365 nm) illumination was negligible. The resulting data (Figure S13b) has been added with a brief discussion in the revised manuscript as follows.

“Moreover, when examined by the colorimetric MTT assay,⁵⁶ the PCL NPs presented minimal cytotoxicity against cells **even upon UV-light (365 nm) illumination for DBTEO photocyclization (Figure S13). According to the minimal cytotoxicity, the PCL NPs hold potential for advanced cell imaging applications based on color-specific photoswitching.**” (p.10, line 21)

Figure S13b. Photo-induced cytotoxicity of the PCL nanoparticles (1 mg mL^{-1} by F-68 concentration) upon exposure to UV-light (365 nm) against RAW264.7 cells. The cell viability was evaluated by the colorimetric MTT assay according to the literature procedure (Mosmann, T., *J. Immunol. Methods* **1983**, 65, 55).

REVIEWERS' COMMENTS:

Reviewer #1 (Remarks to the Author):

After careful reading of all referees' comments, response of the corresponding author and the whole revised manuscript I am convinced that:

1. Authors addressed all comments of all three referees in positive manner.
2. The manuscript is novel (i.e. I disagree with Ref. 3) and I strongly support its acceptance.

Reviewer #2 (Remarks to the Author):

As of my side, the authors have satisfactorily improved the manuscript. I suggest publication in the current form.

Reviewer #3 (Remarks to the Author):

This revision could be accepted for publication.

Response to Reviewers

We are glad that all the reviewers are satisfied with our revised manuscript and support publication. We would like to thank all the reviewers once again for their careful reading of the manuscript and providing many important/constructive/helpful comments.

Reviewer #1 (Remarks to the Author):

After careful reading of all referees' comments, response of the corresponding author and the whole revised manuscript I am convinced that:

1. Authors addressed all comments of all three referees in positive manner.
2. The manuscript is novel (i.e. I disagree with Ref. 3) and I strongly support its acceptance.

Reviewer #2 (Remarks to the Author):

As of my side, the authors have satisfactorily improved the manuscript. I suggest publication in the current form.

Reviewer #3 (Remarks to the Author):

This revision could be accepted for publication.